# Linked Argumentation Graphs for Multidisciplinary Decision Support [note 1]

**DOI:** 10.3390/healthcare11040585

**Published:** 2023-02-15

**Authors:** Liang Xiao, Des Greer

**Affiliations:** 1School of Computer Science, Hubei University of Technology, Wuhan 430068, China; 2School of Electronics, Electrical Engineering and Computer Science, Queen’s University Belfast, Belfast BT7 1NN, UK

**Keywords:** argumentation, argument linking, multiagent systems, clinical decision support, multidisciplinary decisions

## Abstract

Multidisciplinary clinical decision-making has become increasingly important for complex diseases, such as cancers, as medicine has become very specialized. Multiagent systems (MASs) provide a suitable framework to support multidisciplinary decisions. In the past years, a number of agent-oriented approaches have been developed on the basis of argumentation models. However, very limited work has focused, thus far, on systematic support for argumentation in communication among multiple agents spanning various decision sites and holding varying beliefs. There is a need for an appropriate argumentation scheme and identification of recurring styles or patterns of multiagent argument linking to enable versatile multidisciplinary decision applications. We propose, in this paper, a method of linked argumentation graphs and three types of patterns corresponding to scenarios of agents changing the minds of others (argumentation) and their own (belief revision): the collaboration pattern, the negotiation pattern, and the persuasion pattern. This approach is demonstrated using a case study of breast cancer and lifelong recommendations, as the survival rates of diagnosed cancer patients are rising and comorbidity is the norm.

## 1. Introduction

Multidisciplinary clinical decisions are crucial in managing complex diseases or when multiple patient conditions are interleaved. They usually span a diverse range of clinical expertise, time, and locations. Conventionally, clinical decision support systems (CDSSs) could help to interpret clinical data at points of care and assist clinicians in improving adherence to evidence-based guidelines in clinical decision-making. Although each participant may wish to reach conclusions autonomously, they cannot make decisions in isolation but must rather depend on one another. Unfortunately, dynamic interactions among clinicians are inadequately supported in the most current CDSSs.

Agents are computational entities with features of autonomy, concurrency, decentralization, proactiveness, social ability, and flexibility [1]. An agent can make a decision through perceiving its environment and acting upon it to achieve its goal, expressed as maximizing its values, utilities, or benefits. MASs provide an optimal framework to support clinicians both individually at their local sites and, more importantly, collectively across different sites. Agent research interest applied in healthcare has been elevated, with emerging findings reported in recent Special Issues of journals [2,3]. Nevertheless, an agent paradigm alone is insufficient. A range of works modeling individual agents as personal assistants or advisers in addressing various types of medical problems exists [1,4]. Another group of works has concentrated on modeling decision algorithms, e.g., protocols that guide prescription [5] or rules that guide diagnosis [6]. These works and others are limited either in their specific design of agent types for solving specific decision problems or in their centralized nature of decision support. Even though some multiagent models have been proposed, these have mostly remained at the surface of information sharing, while deeper-level joint reasoning has been a lesser concern among multidisciplinary decision makers. For instance, it was pointed out in [7] that clinicians are not well supported in resolving interactions in cases of multimorbidity.

It has been suggested in previous works [8,9,10] that agent-oriented decision-making driven by an appropriate argumentation model is an efficient approach. Argumentation provides a natural means for facilitating the design, implementation, and analysis of sophisticated interaction among rational agents [11]. Ironically, practical applications of argumentation theories to agent-based multidisciplinary decisions in medicine are rather scarce. Despite the fact that arguments compliant with Toulmin’s argumentation model [12,13] or others can be structured in a chain for reasoning, interrelating arguments in support or opposition of each other are usually within closed boundaries, assuming a consistent set of beliefs. Since independent agents have separate argument and belief sets, as well as different goals and constraints, cross-domain inconsistency or domain-specific suboptimization is very likely to arise when these agents collaborate in multidisciplinary decision-making. This is often the case in medicine, as multiple sets of evidence or conflicting interventions deduced from the same evidence in addressing different health issues of the same patient may need to be accommodated. A preferred treatment plan, for instance, may have to be discarded and replanning scheduled when a new patient condition emerges and prevents the plan. Therefore, interagent argument analysis and reasoning at a high level is crucial, and investigation of an appropriate method of argument linking across domains can presumably endow required collective intelligence.

In addition, a notion of pattern is natural to software engineers as a recurring style in addressing similar types of problems and, in this case, argumentation linking in communication. Establishment of such patterns will facilitate future design of multidisciplinary applications and allow developers to meet their own requirements, with a lot of effort saved through building on patterns.

To summarize, this research has two major motivations: (1) There is a lack of systematic support for argumentation in communication among multidisciplinary decision makers. (2) There is a need for identification of patterns in argument linking. To this end, an approach of linked argumentation graphs is proposed to meet requirements we believe to be important to contributing to the current literature, as follows:The approach must be built on the basis of a sound argumentation scheme and suit evidence-based multidisciplinary decision-making. Open standards, e.g., those endorsed by the W3C, and well-recognized paradigms, e.g., multiagents holding beliefs, plans, and decisions, may be applied wherever possible (addressed in Section 3);The approach must identify different types of patterns of linked argumentation graphs across multidisciplinary applications so that reusability is augmented (addressed in Section 4);The approach must solve computational argumentation and also preserve a form that is friendly to clinicians. In doing so, manual efforts from domain experts in argumentation analysis can be saved, and the approach can be adopted in routine clinical practice without extra burden (addressed in Section 5).

## 2. Background and Related Works

### 2.1. Multidisciplinary Decision-Making in Medicine

A multidisciplinary clinical decision involves a group of collaborative specialists who have different areas of expertise and concerns but share the same set of patient data and aim at reaching a consensus of important decisions for the same patient. There are, for example, over 65 decision points in a breast cancer diagnosis and treatment pathway, distributed among GPs, specialist doctors, nursing staff, and even patients themselves. The design of multidisciplinary decision models aimed at multiple parties can be roughly divided into two schools of thought. Firstly, consensus-based decision approaches emphasize the diverse opinions and preferences that decision makers have over alternative solutions. One method [14] suggests that an opinion should be judged in terms of an individual’s rank, level of expertise, relevance, personality, and cognitive style. A commonly agreeable outcome can finally be synthesized, on the basis of these factors, through communication and negotiation. Secondly, evidence-based decision approaches, such as PROforma [10,15], encapsulate the decision rationales of clinical evidence explicitly as arguments for or against decision candidates. Clinicians are expected, as a collaborative group, to contribute together to a final decision compliant with the evidence. However, in those approaches, arguments are understandable only within their own interpretation software, incapable of semantic linking with local electronic health records (EHRs) or other knowledge-based systems, and unsustainable for distribution among responsible parties or seamless integration in practice.

A method of attack graphs for modeling interactions of arguments among various medical professionals is proposed in [16]. Consistency is verified using constraints. In this method, a major constraint is that arguments with greater weights cannot be attacked by arguments with lower weights; otherwise, an inconsistency occurs, and a so-called “bad attack” relationship should be removed. Although cross-domain argumentation can be analyzed using this method, the dynamic and autonomous characteristics of decision-making are largely underestimated. The oversimplified removal operation of “bad attack” relationships does not give full consideration of argumentation from a continuous, evolving perspective, as attacking power can be aggregated, and even a strong, present argument could be defeasible in a future extension. Furthermore, attacking is not the only type of interaction among decision makers. Most importantly, the arguments in this work are simply individual pieces of advice from medical professionals, and there is a lack of formal argument representation using well-established sources such as clinical guidelines.

A goal-oriented methodology is suggested for patients with multimorbidity in [7]. Computer-interpretable guidelines (CIGs) are specified as goal-oriented modules; hence, goals can support reasoning and direct selection of alternative plans based on argumentation. While new goals for managing patients’ progressing health states arise, emerging inconsistencies among guideline recommendations can be mitigated. However, this method employs a centralized “Controller” component for detecting inconsistencies in goal forests and generating alternative nonconflicted management plans. This does not suit multiple clinicians interacting autonomously in their routine practices in a distributed environment. In addition, the targeted inconsistencies are limited to either the starting/stopping of medications that belong to the same class hierarchy or to opposing physiological-effect goals, rather than more general inconsistencies that appear in medical interactions.

### 2.2. Argumentation Theories and Representations

The issue-based information system (IBIS) [17] is capable of capturing argumentation of alternative solutions around a question to solve. Its key notations include questions, ideas, pros, and cons. Arguments are expressed in natural-language sentences and linked in a diagrammatic rather than a computational manner. The IBIS underpins the logic of argument (LA) [18] as applied in medicine and the agent-facilitated crowd discussion platform of D-Agree [19].

The argument interchange format (AIF) [20] was proposed for semantically rich and computationally enabled representation of arguments. An AIF core ontology and its extensions are defined to help to construct argument networks. A world wide argument web (WWAW) [21] was put forward on top of the AIF, with an envisioned future that arguments on the WWW can conveniently interconnect on a very large scale [22]. AIFdb and ArgNav [23] can be used for storage, navigation, and analysis of arguments.

DebateGraph was developed for construction and visualization of arguments. It is, however, incapable of engaging multiple stakeholders and interlinking arguments semantically, either within their own environments or with the rest of the web. The value of formal decision models has not yet been exploited and the semantics of arguments not explicitly specified, so they lack automated argumentation capability [11].

For many stakeholders becoming involved in decision-making, their diverse viewpoints as well as scientific evidence should be accommodated in coping with critical issues in a safe and sound manner. Dialogical argument interactions were studied in [24] to address sociopolitical disputation. That study proposed an institutional framework in which polemic viewpoints and controversies could be traced to lower-level facts and events. In that hierarchy, the authors suggested that dialogues of various types would need to be interleaved as a way of modeling controversies.

The argumentation theories and tools mentioned above, as well as other such developments, offer a versatile and effective foundation for decision-making and reasoning under uncertainty [25]. In terms of representation, the triple structure of a resource description framework (RDF) is a natural candidate for expressing propositions formally [26]. It has been demonstrated that Toulmin’s argument model can be defined using a generic ontology [27], which could later guide representation and interpretation of arguments in interchangeable RDF structures. Methods have been developed to maintain semantic relationships among clinical concepts [28] and acquire semantically enriched data for clinical queries [29] or storage [30] and toward clinical recommendations using semantic web rule language (SWRL) [31]. In comparison, knowledge graphs can be regarded as networks of concepts with semantic relationships between them. They can be represented as RDF graphs or collections of triples: <*s*, *p*, *o*>. A triple consists of a subject, *s*; a predicate, *p*; and an object, *o*, making a statement about a relationship, *p*, between *s* and *o*, or a link between two nodes in the RDF graph. A number of knowledge graphs have been constructed in medicine [32,33,34]. Usually, a data schema or a knowledge graph model level is designed, and then data entities are extracted from sources, followed by a process of entity alignment and mapping. In addition, linked open data establishes semantic links among various data sources, and this may help to promote user-oriented recommendations [35]. Despite advances in knowledge graphs and linked open data with capabilities in semantic queries and so on, they are insufficient in capturing explicit argumentation structures. Thus, they are not natural solutions for decision support: in particular, facilitating explanations of decision rationale. Nevertheless, an RDF-based representation of argumentation can ease later integration with these current facilities, enabling straightforward recommendation generation, understanding, visualization, and exploration in future investigation.

### 2.3. Multiagent Argumentation

The MAS paradigm provides an effective framework for coordinating local knowledge, computation, and reasoning in a global and uncertain yet dynamic environment. This has led to consideration of MAS as a suitable candidate in the design of multidisciplinary decision support. In this context, two important domains of techniques provide a substantial foundation for agent reasoning. Argumentation theories [36] are concerned with agents changing the minds of others, and belief revision models [37] are concerned with agents changing their own minds. They are closely related to each other, since good decisions rely heavily on both adequate plausible beliefs and rationale argumentation. Arguably, an agent argumentation structure built on the basis of belief would provide a more useful means for analysis of argument interrelationships. In the influential AGM paradigm [38] and its refinement [39] beliefs are characterized as sets of propositions, and belief expansion, contraction, and revision are concerned at the time that an agent receives information inconsistent with the present epistemic state. In data-oriented belief revision [40] belief selection is based on incoming data and their properties, e.g., source credibility, motivation, and emotional features, as possible cognitive reasons to believe them or not. Nevertheless, these methods are centered on an individual perspective of reasoning regarding agent interaction with an entire outside world. Therefore, literally no support has been provided for analysis of agent–agent argument interaction, and no common pattern has been identified or represented to inform the reuse of such design in various communication scenarios.

Graph-based transformation [41] has been proposed as a means through which multiple agents can exchange their locally maintained knowledge. In particular, it was illustrated that RDF graphs possessed with different agents could be queried and new pieces of knowledge produced when agents needed to cooperate. Nevertheless, any transformation is limited by the data nodes and edge attributes through which a knowledge representation graph may be altered. In this sense, knowledge remains at the level of beliefs that an agent holds, without any reasoning on when or how such a transformation should be made.

In the setting of multidisciplinary decisions, our view is that any type of revision of belief in agent communication would be overcomplicated when tangled with changing plans or decisions, essentially. It may be more viable to regard multidisciplinary decisions as a process of multiple agents collectively selecting what propositions to put forward while retaining their local autonomy at the belief level. Therefore, a clear distinction may be drawn between the notions of belief and proposition. As such, a belief layer can be regarded as basic and locally effective to its holding agent. For example, patient symptoms are separate from a derived proposition layer containing mutually acceptable treatment plans. It is at the proposition level that agents interact explicitly and argue amongst one another. Beliefs, possibly but not necessarily associated with strengths upon adoption, can be shared among agents. An important advantage of such a design is the endurance of inevitable inconsistent agent states among multidisciplinary decision makers. In fact, clinicians are most likely to hold different sets of beliefs or views toward patients, but that does not necessarily draw immediate attention unless contradictory interventions and the like result from them. Minimizing the complexity of an agent argumentation scheme helps so long as the possibility of inconsistency is accepted and the point where it is addressed is postponed until a consensus amongst multiple plausible but contradictive actions is reached.

## 3. Materials and Methods: A Clinical Argumentation Scheme and Multiagent Argumentation Graphs

A clinical argumentation scheme is proposed here, as shown in Figure 1. The key elements include the decision candidate (C), a clinical option such as surgery, chemotherapy, or an endocrine therapy of tamoxifen; the patient clinical statement (CS), a clinical expression such as the presence of symptoms, signs, and examination results of a patient; and the argument (A), a proposition that argues about a candidate, supported by clinical evidence (E). The argumentation holds the whole structure together; a decision candidate (C) is asserted through a patient clinical statement (CS) for reasons given in an argument (A), unless some opposing views are acknowledged as rebuttal (R). The rebuttal element may or may not be present in this scheme because one may specify an opposing view as an argument against a decision candidate with the same effect. An argument can have a support type (S) of “for” or “against” and a weight (W) indicating its strength. This scheme is an extension of Toulmin’s argument model [12].

An **argumentation graph** can be constructed using this scheme in such a way that an agent can aggregate every case of clinical statements, arguments, decision candidates, etc., available of a given domain and produce an interconnected graph of knowledge. We consider **beliefs**, **plans**, and **decisions** as key agent components being mapped from the graph elements. The set of clinical statements (CS_i_) corresponds to agent belief, that of candidates (C_i1_, C_i2_… C_in_) corresponds to agent plans, and the preferred candidate set (C_ix_) corresponds to the agent decisions, as shown in Figure 2.

Only valid clinical statements shall be established as beliefs; this enables evaluation of the propositions present in arguments to be either true or false, as shown in Figure 2(a);All candidates shall be established as the plans available, and they may be supported or opposed by arguments in varying degrees, as shown in Figure 2(b);The best candidate shall be recommended as the final decision via aggregating the overall supporting/opposing argument weights of the candidates and ranking them based on their strengths, shown in Figure 2(c).

Originating from evidence, arguments are the driving force of domain-specific decision-making. Although cross-domain agents do not share the same sets of arguments or beliefs, they do continuously update what they should believe or how they should act in their interaction, thus linking their originally independent argumentation graphs together. The generic agent interactions in Figure 2 will be instantiated later, while various patterns of linking are discussed in Section 4 to address various types of problems.

In illustrating representation of argumentation graphs in compliance with the above scheme, a triple-assessment case study was introduced as our evidence source. Triple assessment is a common procedure in the National Health Service of the UK for women suspected to have breast cancer and referred to specialized breast units. Patients may be presented by their GPs or following routine breast screening, i.e., the NHS Breast Screening Programme (in England) or the Breast Test Wales Screening Programme. In both situations, it is the best practice to carry out, in the breast unit, a “same day” clinic for evaluating the grade and spread of cancer, if any, or a “triple” assessment: clinical and genetic risk assessment, imaging assessment, and pathology assessment. A multidisciplinary team should be constituted by healthcare professionals with different areas of expertise, including GPs, nurses, radiologists, pathologists, oncologists, surgeons, and so on. An optimum care pathway should be selected for each patient individually. This is used as a running example for demonstrating the method throughout this paper and will be extended progressively in later sections. A part of the evidence for imaging assessment from the Royal College of Surgeons of England is summarized in Figure 3.

In alignment with clinical evidence, a general three-step process of (1) **candidate elicitation**, (2) **argument identification and grouping**, and (3) **statement construction** is employed for elicitation of argumentation scheme elements. The details of this process can be found in [43] as part of our previous work and are omitted for the conciseness of this description. This process results in the construction of a fragment of an argumentation graph and its RDF representation, as shown in Figure 4. 

As such, a full argumentation graph can be built using the evidence; a small portion of such a graph is shown in Figure 5. This example has a total number of 45 argument nodes with a uniformly layered structure. For instance, one clinical argument says that a patient being pregnant (CS) is an argument (A) against (S) a mammogram (C) for the reason of potential radiation risk to the fetus (E). A primary RDF triple is represented as <statement, argument-against, decision-candidate of mammogram>. A secondary RDF triple is represented for the statement as <patient, currentlyPregnant, true>.

## 4. Three Patterns of Linked Argumentation Graphs

On the basis of the multiagent argumentation graph, we will discuss next three types of linked argumentation graph specified for multidisciplinary decision support. To assist with understanding, a fictional scenario is described below so that the reader can easily relate these graphs to scenarios where agents are attempting to change the minds of other agents and/or their own.

**Collaboration Pattern** (changing the minds of others in what to believe): agents share beliefs and support each other in selecting the best choices available (between GPs and surgeon agents, as below, detailed in Section 4.1).**Negotiation Pattern** (changing the minds of others in how to act): some but not all agents must change the best choices they have in mind, and every agent may maintain its original beliefs (between breast-cancer and depression-manager agents, detailed in Section 4.2).**Persuasion Pattern** (changing the minds of others in what to believe in an attempt to change their minds in how to act): agents may or may not maintain the best choices they have in mind but must update their original beliefs (between clinicians and patient agents, detailed in Section 4.3).


*Mary is a 37-year-old, recently pregnant woman with a family history of breast cancer. She has just been urgently referred from her GP to a surgeon due to the finding of a discrete, hard lump; nipple distortion; and other symptoms. These are informed to the surgeon in collaboration, and later, a further radiology investigation using an ultrasound or a mammogram is recommended (collaboration pattern).*



*While Mary is diagnosed with breast cancer, an adjuvant therapy of tamoxifen is prescribed. Sadly, she develops depression after some time due to the breast cancer, and then fluoxetine is prescribed as an antidepressant. Unfortunately, a drug–drug interaction exists between tamoxifen and fluoxetine. Mary’s clinical teams from both breast cancer and depression decide to keep the tamoxifen and switch the fluoxetine to sertraline to resolve the issue following negotiation (negotiation pattern).*



*In considering chemotherapy or tamoxifen as alternative neoadjuvant therapies, clinicians and Mary discuss the pros and cons. Prior to this, chemotherapy was the original clinical recommendation, but the clinicians and Mary are worried that it has far more side effects than what the patient can possibly bear. Later, it is understood that Mary is strongly against hair loss, moderately against fatigue, and slightly against loss of appetite, all of which are side effects of chemotherapy. Taking into account both clinical evidence and patient preference, the clinicians are eventually convinced that tamoxifen is a better choice than chemotherapy (persuasion pattern).*


### 4.1. Collaboration Pattern

In the collaboration pattern, agents collaborate and support each other, and their belief and argument sets can coexist without causing any contradiction: e.g., experts from multiple disciplines joining together to reach a diagnosis decision. Cross-domain agent communication can help agents to share new beliefs and support argumentation in other domains, directly or indirectly. As an example, shown in Figure 6, with both CS1 and C1 established in Agent1′s argumentation, this supports Agent2 to lead to its conclusion of C2. It is vital that clinical findings and other data available to Agent1 are also available to Agent2 in the form of belief sharing. The above process goes on iteratively to eventually reach a final diagnosis. Two subtypes of the pattern can be distinguished: direct argumentation support between Agent1 and Agent2, Agent2 and Agent3, etc., and indirect argumentation support between Agent1 and Agent3, Agent2 and Agent4, etc. It is worth noting that other, unselected decision candidates that may exist are omitted in the linking of the argumentation graphs for simplicity.

In our case study, upon a patient encounter, a GP agent found that a patient was over 30 years old, with a discrete, hard lump and nipple distortion. Several arguments supported an urgent 2-week referral of this patient [44]. On receipt of the referral, the surgeon agent adopted the above findings as their own belief. Following examination of a family history that indicated higher-than-population risk, consideration of the nature of the lump, and looking at other symptoms, the agent argued among the existing plan options of further investigation, discharge, and managing the patient. In a typical case, the agent would decide to choose further investigation as its decision, since the patient condition was suspicious and breast cancer could not be ruled out at that stage. The patient was then referred to a radiologist agent. The same process occurred between the radiologist agent and the pathologist agent, etc., with each stage of the accumulation of agent beliefs toward the patient, sharable in later stages.

As a typical scenario, the pattern of linked argumentation graphs is as instantiated in Figure 7. The multiagent argumentation graph in Section 3 is referred to, and some key arguments for the GP and the radiologist agent are provided in the figure. The agent communication, in which text- or multimedia-based message passing could be applied, provides a common belief update mechanism and also fulfills the actual linking.

Two special extensions of this pattern can be specified in Figure 8 as (a) “All for one” and (b) “One for all”. In the first case, a conclusion is drawn from a major clinical statement, which in turn is a combination of multiple other, independent conclusions. In the second case, a major conclusion is drawn and multiple clinical statements are inferred from it, which in turn leads to independent conclusions. Thus, an important argument may either lead to or be inferred from the conjunction of many others.

### 4.2. Negotiation Pattern

In the negotiation pattern, agents work together, and their plans must be negotiated to avoid adverse effects or suboptimal global medical effects, e.g., how multiple treatment plans for managing a patient with comorbidity cannot coexist. The negotiation mechanism of this pattern is shown in Figure 9.

Prior to this pattern coming into function, agents would independently lead to their preferred plans. Unfortunately, they would contradict each other. This may be due to the fact that two plans are of the same type and have opposite goals or actions (such as drug interactions) for the patient. Both agents would initiate negotiation in the hope of each maintaining their own best choice (Step1 in Figure 9). However, the negotiation would be up to their relative strengths or, say, the extent of persistence each agent holds toward its plan; one of the agents would have to abandon its preferred choice and replan. Here, the established argument structure from the other side can be regarded as a new, independent argument against a current plan, and both plans would be undermined (Step2 in Figure 9; both 2-1 and 2-2 take place). The one more vulnerable to the weakening effect of this new argument would have to accept the negotiation and switch to an alternative option that is less preferable but with a near-equivalent effect (Step3 in Figure 9; either 3-1 or 3-2 takes place). In the end, this would be a mutually agreeable solution for both parties and beneficial to the patient on the whole.

As a demonstration, our case study includes an extraction of evidence, shown in Figure 10. The decisions are related to the care of breast cancer patients with comorbidity of depression as well as an overall concern of bone health.

In Figure 11, an instantiation of the negotiation pattern for a case study in a temporally evolving manner is shown. The agent structure is simplified to show decisions only, in this case, the chosen prescriptions. Additionally, we define below a set of health-concern characteristics as enabling facilities for negotiation.

**Persistence**: A measure of the extent to which one wishes to stick to a candidate. It is proportional to the overall weight that a candidate is granted and could be weakened through a cross-domain argument against it.

**Relevance**: A disease or a treatment is of relevance if it is a risk factor for a given health concern: e.g., when a patient diagnosed with breast cancer and prescribed an antidepressant displays two independently relevant factors to bone health.

**Importance**: A measure of the risk factors of relevance to a given health concern: e.g., when bone health has an importance level of 3, given that one is diagnosed with breast cancer with a prescription of SSRIs and an excessive consumption of alcohol.

**Critique**: A given health concern becomes critical if its importance is above a threshold value; e.g., bone health becomes critical once its importance is over 3.

Suppose a patient is diagnosed with breast cancer in the first instance, with an adjuvant therapy of tamoxifen prescribed (T1 in Figure 11). After some time, she develops depression due to the breast cancer, and then fluoxetine is prescribed as an antidepressant (T2). It was indicated in [46] that a drug–drug interaction exists between tamoxifen and fluoxetine, with an enzyme of cytochrome P450 2D6 (CYP2D6) playing a central role in tamifoxen’s efficacy, while some SSRIs, including fluoxetine, are known to inhibit CYP2D6. Suppose tamoxifen is insisted to a greater extent; fluoxetine may then be switched to sertraline (T3), indicated with a relatively higher persistence of a breast cancer agent for tamoxifen over a depression manager agent for fluoxetine). Sertraline is another type of SSRI, with lesser degrees of inhibition. In contrast, if one insists that mental health problems should be treated well to guarantee effective treatment of breast cancer and supposes the patient is a postmenopausal woman, it is justified that tamoxifen may be switched to an aromatase inhibitor of anastrozole to resolve the conflict. In either case, both the presence of breast cancer and prescribing SSRIs increase the risk of bone health. If the patient’s lifestyle is negatively affected due to depression and this causes her nutritional deficiencies, this would further worsen the situation of bone health, with a critical alarm fired (T4, critique = yes). As a result, vitamin D3 would be prescribed as an additional decision to mitigate the loss of nutrition, and the critical level would be back to normal for the time being (T5).

### 4.3. Persuasion Pattern

In the persuasion pattern, an agent may be persuaded to reconsider their best choice of plans because they are convinced by another agent, e.g., that a preferred therapy has to be discarded, as it would bring far more side effects against the patient’s preference than would others. As a persuasive power exists, one or more agents may change their views (“persuaded to believe”) or actions (“persuaded to do”). Two subtypes of the pattern, as shown in Figure 12, are distinguished; a decision of one domain may be directly attacked by a new conclusion from another domain (a rebutting defeater) or indirectly undermined by it as one of its previously supporting arguments is invalidated (an undercutting defeater). The two domains here may be separate disease problems or different perspectives on the same disease as managed by different agents. A major difference between the persuasion pattern and the negotiation pattern is that some agents must change their choices in negotiation, and persuasion may either succeed or fail. This implies that persuasive or convincing power may be insufficient to change one’s position; e.g., taking into account the side effects of a treatment may or may not affect the original clinical choice. In the negotiation pattern, either side could make the concession, and in the persuasion pattern, it must be a certain side if it does indeed happen. In this study, we focus on patient preference as the persuasion force, as patients are themselves important participants involved in multidisciplinary decisions. Clinicians might be influenced by patient attitudes and persuaded to choose alternative treatment options if appropriate. This pattern can be extended and incorporate other persuasion forces in the future.

The application of this pattern can be demonstrated with our case study of breast cancer treatment, as a previously considered best choice no longer held due to its undesired side effects and the patient preference. The clinical evidence of NICE NG101 [47] and the NHS health guidelines [48] were referred to. Four different types of therapies available for selection, as well as their common side effects, are shown in Table 1. Additionally, review data from a selection of 600 patients was collected from the community site Patients Like Me [49]. To simplify matters, we regarded the frequencies of side effects following a treatment as being reflected in the number of their quotes out of the total reviews per therapy. Statistic calculation and analysis gave us a rough estimation of the association between side effects and therapies, shown in the frequency column of Table 1. Sample reviews for chemotherapy are also given. Please note that the established association may be clinically biased due to the limitation of the sample. Nevertheless, this gives a foundation for building evidence for side-effect arguments. In the preliminary study, side-effect weight was measured based on frequency and computed as individuals that experienced the side-effect consequence divided by the total number of patients coexposed to the side effect through the treatment involved. Frequency has been used as a core element to build evidence of harmful drug interaction, as recommended by W3C [50]. Seriousness is another element in that model, and its measurement via deeper analysis of review data will be carried out in our future work to produce more accurate weight values.

An example of application of the persuasion pattern is shown in Figure 13. An argumentation graph applying the clinical argumentation scheme is shown in the lower part. That scheme is extended here to yield a patient-preference argumentation scheme, with its associated argumentation graph shown in the upper part. The original scheme’s main structure was largely maintained, whereas a clinical statement (CS) became a preference statement (PS) about patient expression toward side effects: argument (A) argued about a candidate (C) treatment of a side effect, with a support type (S) of “against” and a weight (W) indicated through its frequency. In establishing the PS, patients were prompted with a five-point scale to assign marks between 0 and 5: A mark of 5 indicated feeling strongly against a side-effect impact, 3 indicated feeling moderately against an effect, 1 indicated feeling slightly against an effect, with 0 indicating no difference at all. This enabled personalized decisions on the basis of individual patient assessment.

In Figure 13, a rebuttal (R) relationship is established between the schemes regarding the decision candidates relating to chemotherapy. The calculation of preference marks is shown through multiplication of each patient-assigned side-effect mark based on its weight and aggregating the weighted scores. In comparing chemotherapy with an alternative therapy of tamoxifen, chances are that chemotherapy may be recommended by clinicians in the first instance, and a joint decision with a patient may suggest the opposite. Chemotherapy may become less preferred over tamoxifen due to the fact that the former has far more side effects against patient preferences than the latter (assuming a relatively insignificant clinical difference between the two options). This would lead to a successful persuasion.

A special extension of this pattern can be specified, as in Figure 14, as a way of selecting mutually exclusive plan sets. In four different domains, Plan1a and Plan3a can be selected together; alternatively, Plan2a and Plan4a can be together, but all four plans cannot be selected together. If one set is determined, the other sets must be abandoned.

## 5. Results: Implementation and Demonstration

An argumentation engine was developed to support runtime interpretation and linking of RDF-based argumentation graphs. Algorithm 1 shows its main algorithm; it first builds an RDF model using Jena (line 1); then collects decision candidates, arguments, and statements from the model (lines 2–7); and finally justifies the arguments, aggregates the total weights of the candidates, and returns them in a ranked order (lines 8–12). An argument linking is carried out on request prior to the outcome returned (lines 10–12). The result is eventually available to the decision support interface.
**Algorithm 1** The algorithm used in the engine for argument interpretation and linking.Argumentation-Engine()1*model* = Model-Factory-Read()2*Candidates* = Selector(*model, Property.decision*)3*Map < Candidate, Arguments > map*4**for***i* = 0 **to**
*Candidates. length*
**by** 15  *Arguments* = Selector(*Candidates*[*i*]*,* “*consist*-*of* ”)6  **for**
*j* = 0 **to**
*Arguments. length*
**by** 17    *Statements* = Selector(*Arguments*[*j*]*,* “*include*”)8    *A*[*Argument*][*weight*] = Verify-Argument(*Statements*)9  *map. put*(*Candidate*[*i*]*,*Collect(*A*[*Argument*]*,*Calculate(*A*[*Arguments*][*weight*])))10   **if** (*linking-request*)11   Link-Argument(*map*, *linking-request. getMap*())12**return** 
*map*Verify-Argument(*Statements*)1*OrStatements* = *Split*(*Statements*, “*or*”)2*i* = *j* = 03**for***k* =0 **to**
*OrStatements*. *length*
**by** 14  AndStatements = Split(OrStatement[*k*], “and”)5  **for**
*m* =0 **to**
*AndStatements*. *length*
**by** 16    **if** (Judge-Patient-Data(*AndStatements*[*m*]))7      *j* = *j* + 18  **if** (*j* == *m*)9     *i* = *i* + 110     statement = OrStatement[*k*]11     Break12**if** (*i* > 0)13   *A*[*Argument*][*weight*] =Add(*statement*,Selector(*statement*, “*weight*”))14**return***A*[Argument][weight]Link-Argument(*map*, *map’*) 1**for***i* = 0 **to**
*map. size*() **by** 12 **for** *j* = 0 **to** *map’. size*() **by** 13  **if** (*map. get*(*Candidate*[*i*])*.equals*(*map’. get*(*Candidate*[*j*])))4   *map*. *merge*(*Candidate*[*j*], *map’. get* (*A*[*Argument*]),Calculate(*map. get* (*A*[*Arguments*][*weight*]), *map’. get*(*A*[*Arguments*][*weight*])))5**return** 
*map*

A verify-argument function is defined and invoked in the main algorithm in line 8. It takes in a composite argument statement and splits it into components separated by the “OR” keyword. Each component is further split into subcomponents separated only by the “AND” keyword. Two iterations are used to verify an argument: the whole argument is valid given that at least one of its “OR” components is judged valid, then in turn all its “AND” subcomponents being successfully judged valid. A link-argument function is defined and invoked in the main algorithm in line 11. It checks, recursively, whether two equivalent candidates exist in the argumentation structures held by two agents. Upon the detection of such, the two collections of verified arguments are merged for that same decision candidate, as are the overall supporting weights. This is exemplified in patient-preference argumentation and clinical argumentation in the persuasion pattern for chemotherapy and tamoxifen. The merging operation can be assigned different mechanisms to achieve the desired effects, i.e., to accept new beliefs in the collaboration pattern and accept new arguments in the negotiation pattern.

A prototype system was developed, on the basis of the algorithm, to demonstrate the feasibility of the approach. It is shown in Figure 15: a multidisciplinary decision support interface for breast cancer. At this particular point, an agent would present its belief–plan–decision structure. The belief and plan parts are summarized at the top (Figure 15a) and the details on decisions in the middle left (Figure 15b) for consideration. Figure 15a shows a summary box of the previous execution outcomes of the **collaboration pattern**, with belief sharing between this agent and others. In the bottom (Figure 15e), another summary box of the outcome of the **negotiation pattern** is presented, with another agent concurrently managing a comorbidity condition and changing its plan due to its acceptance of the current agent’s request of negotiation.

The main interface shows the patient and clinicians making decisions together, and the **persuasion pattern** of linked argumentation graphs is applied. Two of the treatment candidates are presented in particular and ranked based on their aggregated weights (tamoxifen: −0.489 and chemotherapy: −0.77). This alters the original clinical consideration (chemotherapy: 2 and tamoxifen: 0) prior to persuasion. The algorithm shown in Algorithm 1 is applied to yield the results here. The decision makers can freely adjust the relative importance of clinical argumentation and preference argumentation on the fly via assigning a calculation ratio between two parties (Figure 15b). The clinical evidence, as well as patient reviews, is retrieved and presented for explanation in the due course of decision support (Figure 15c). A graphical view of linked arguments is available for user navigation of the decision candidates, their pros and cons from both clinical and patient perspectives, and their persuasion relationship (Figure 15d). Effectively, the two parts of the argumentation are linked, and since the persuasion power is strong enough, clinicians are convinced that the original clinical recommendation of chemotherapy should be switched to tamoxifen. Eventually, the linked argumentation graphs serve as a comprehensive decision aid patients and clinicians together.

## 6. Experiments and Evaluation

We carried out an empirical experiment to evaluate our approach. The prominent features of this experiment and its major differences from those [8] we conducted previously include the fact that decision makers were grouped together rather than as individuals. In addition, the cases under investigation required interaction of decision makers so that they might support each other, concede to avoid harm in addressing comorbidity, and concern patient preferences.

### 6.1. The Preparation of Decision Support Systems for Comparison

Three CDSSs were prepared prior to carrying out the experiments. The prototype system described in Section 5 was used for comparison, supporting multidisciplinary decision-making via linked argumentation graphs (called a LAG-CDSS). In addition, a conventional decision support system was developed using PROforma (called a P-CDSS), as well as a basic clinical decision-making system with no recommendation at all (called a B-CDMS). The interfaces of the two systems are shown in Figure 16, the data collection interface (a) being shown in both systems alongside the resulting output (b).

Both CDSSs shared the same clinical guidelines and guided decision makers in the decision processes of collecting clinical data: recommending decision options in ranked lists, prompting clinical actions to commit, etc. Clinicians from the LAG-CDSS group might have accepted (or rejected) the decision recommendations ranked as the top options in Figure 15b following the examination of overall context presented in the interface. Clinicians from the P-CDSS group had access to independent decision support facilities without being linked together. From the perspective of a physician of breast cancer, she could not directly collaborate and share beliefs with the GP or other clinicians via enactment of the collaboration pattern. Similarly, she could not directly negotiate for changing plans with other physicians, who manage comorbidity conditions, via enactment of the negotiation pattern, or persuaded to rerank her options using patient preferences via enactment of the persuasion pattern. The components of (a), (d), (e), and part of (b) of the prototype system in Figure 15 were unavailable. Instead, offline interaction was necessary for joint decision-making. Clinicians from the B-CDMS group were asked to provide their decisions without any support in both individual recommendation and group interaction.

As multidisciplinary decisions were concerned for each patient case, separate decision interfaces were activated for individual decision makers in a distributed manner wherever possible. This happened not just for clinicians working on different diseases, i.e., physicians working on management of breast cancer and depression, but also for those working on the same disease but playing different roles. The temporal dependencies between decisions determined participant involvement in patient cases.

### 6.2. Research Hypotheses

Four metrics were set out for the evaluation of three systems, namely, accuracy, time, satisfaction, and learning, falling into the categories of productivity, process, and perception. Such a categorization was recommended in [51,52] for metrics of CDSSs. The categories, their metrics, and their descriptions were given in Table 2.

Our hypothesis was that the prototype system built with the new approach would outperform existing CDSSs or basic clinical decision-making systems without support in the above metrics. Precisely, four hypotheses were defined as follows.

**Hypothesis** **1** **(H1).**
*The LAG-CDSS supports the generation of more accurate group level decision outcomes than do the P-CDSS and the B-CDMS.*


**Hypothesis** **2** **(H2).**
*The LAG-CDSS requires less decision-making time than do the P-CDSS and the B-CDMS.*


**Hypothesis** **3** **(H3).**
*The LAG-CDSS makes clinicians and patients feel more satisfied with it over the decision-making processes and outcomes than do the P-CDSS and the B-CDMS.*


**Hypothesis** **4** **(H4).**
*The LAG-CDSS enables clinicians to learn more from it than do the P-CDSS and the B-CDMS.*


### 6.3. Experimental Settings

Firstly, we prepared a set of 30 patient cases, recorded in the past 10 years, from a major national Grade-A tertiary hospital in Wuhan City. The process followed a data anonymization protocol regulated by the hospital and was assisted by colleagues from the Breast & Thyroid Surgery Department.

Then, 60 postgraduate medical college students were recruited to join the experiments. They had all been trained with corresponding background medical knowledge and up to one year of medical practice experience, and so were considered junior clinicians. Statistical tests were carried out, and no significant difference was found among them. The students were split into 15 groups, with four members in each: one playing the role of a GP, one as a breast cancer physician, one as a depression physician, and the last one as a patient. These groups were assigned evenly and randomly to one of the three systems, LAG-CDSS, P-CDSS, or B-CDMS, resulting in five groups in each.

Finally, a total number of 10 patient cases were assigned to each system for running through decision-making processes. Each group worked with its designated system on the assigned cases. The decision outcomes and the time spent on patient cases were automatically recorded in these systems. Questionnaires were handed out afterward, and participants were asked to rate their feelings toward satisfaction and learning. Every measure was employed to ensure that all groups successfully completed the experiments.

Accuracy was directly measured, via observation of the decision-making processes, as the total accurate decision outcomes among 10 patient cases per system. Time was measured via observation of the beginning and end points of the entire process and calculation of duration as the time spent per case. Satisfaction and learning were measured using questionnaires with a five-point Likert Scale. The questions were centered on the most prominent features of the prototype system: whether decision makers were satisfied with or could learn from the support for collaboration or negotiation among decision partners, the presenting of both clinical arguments and patient-preference arguments, the functionalities of recommendations and explanations, etc.

### 6.4. Results of the Evaluation and Analysis

We collected data on accuracy of decision outcomes and time taken for decision processes, as well as group feedback on satisfaction and learning. An analysis of variance (ANOVA) was used as a tool to determine if the difference between group means was statistically significant. In particular, two *p*-values were used to test the previously defined hypotheses. The first assumed the null hypothesis of no statistical difference between LAG-CDSS and P-CDSS; the second assumed the null hypothesis of no statistical difference between LAG-CDSS and B-CDMS. A *p*-value less than 0.05 would allow us to reject the null hypothesis, and support the alternative hypothesis (statistical difference). The results are shown in Table 3, and the mean values of the metrics and their comparison are in Figure 17.

It can be seen from Table 3 that three hypotheses, **H1**, **H2,** and **H4,** were confirmed using the first statistical measurement (*p*-value1 < 0.05), and all hypotheses, **H1**–**H4**, were confirmed using the second statistical measurement (*p*-value2 < 0.05). The results revealed a stronger association of accuracy, time, and learning with the new approach than with conventional CDSSs or with decision makers on their own, with no support. As for **H3**, the results indicate that there was no statistically significant difference (*p* = 0.2675) between the LAG-CDSS and the P-CDSS in terms of satisfaction, though the mean value of the former was greater than the latter. We believe that two reasons contributed to this: (1) Some participants prefer more flexibility, autonomy, and face-to-face communication as working groups. A short interview following the analysis of the questionnaires revealed that some groups felt more comfortable working independently and discussed mutual issues as partners only when necessary. This way, they sensed more independence and confidence; (2) Although the inclusion of patient preference in decision-making is considered a manner of increasing overall satisfaction and a major advantage of the LAG-CDSS, the chosen participants could not fully represent patients in reality. This led to the elimination of such advantages via the overwhelming clinical opinions in the experimental setting. The involvement of patients in the real world will be part of future studies allowing a more extensive evaluation.

Most of the participants agreed that they learned something from this experience. Many recognized that new, innovative techniques might improve the efficiency of communication, the understanding of the current status of cases, and the way support groups work together. Some were excited about the potentials of such tools and felt keen to become engaged in an integrated decision-support environment in practice.

Thus, the use of the linked argumentation graphs approach resulted in statistical significance in accuracy, time, and learning for multidisciplinary decision-making. The mean value of satisfaction using the CDSS built with the new approach was higher than that of the system built with the conventional approach, but did not show a statistically significant improvement. These favorable results align with our primitive research goal and suggest that the proposed methods could be a substantial contribution to the current CDSS literature.

## 7. Discussion and Conclusions

In this paper, we proposed an approach of linked argumentation graphs for multidisciplinary decision support. First, we put forward a clinical argumentation scheme on the basis of which multiagent argumentation graphs could be constructed with the key agent components of beliefs, plans, and decisions. Then, the representation of argumentation graphs was illustrated using a triple-assessment case study. After that, we discussed three types of linked argumentation graphs—the collaboration pattern, the negotiation pattern, and the persuasion pattern—in accord with situations when agents attempted to change the minds of other agents and/or their own. A case study was enriched extensively for illustration of these scenarios. Finally, we presented the design of an algorithm used in our engine for argument interpretation and linking and the development of a prototype multidisciplinary decision-support system with the enactments of three types of patterns. The prototype system was evaluated against a conventional CDSS and a decision-making system without support, demonstrating improvements in the metrics of accuracy, time, satisfaction, and learning.

In our previous work, an agent-oriented framework was developed to deliver decision support in compliance with guidelines [10]. Additionally, another work was developed to deliver the extraction of patient sentimental opinions in alignment with an ontology [53]. We are, naturally, reaching a point of investigating a systematic approach toward cross-domain argument interaction. This should take into account not only integrating evidence semantically but, more importantly, the patterns that recur among agents in collaboration, negotiation, persuasion, etc., in various multidisciplinary decision scenarios. We argue that the design of the generic clinical argumentation scheme and its related multiagent argumentation graphs provides a solid foundation and plays a key role in categorization of situations where agents attempt to change the minds of other agents and/or their own. In summary, this work has three major contributions, as follows:*Multidisciplinary decision support with a solid theoretical foundation as well as practical applicability*: the extension of Toulmin’s model toward clinical argumentation provided us a solid foundation theoretically, and, when mapped to multiagent argumentation graphs, it became applicable practically;*Reusability of the design for future multidisciplinary decision support systems via identifying recurring patterns*: three types of patterns of linked argumentation graphs across multidisciplinary applications were identified and demonstrated using corresponding scenarios;*Both human-understandable and machine-executable*: the graphical representation supported intuitive user navigation and reasoning, and the supporting engine enabled argument interpretation and linking.

These contributions, at the same time, answer the requirements raised in the end of Section 1.

Despite being simple and straightforward, this argumentation scheme supports three useful patterns for argument interaction. We will look further into its generality and coverage across clinical scenarios. One avenue is to explore a variety of argumentation schemes as suggested by Walton [54], that could guide building of argument networks from the ontological level down to intermediate node types and, finally, actual arguments [21]. Furthermore, we wish to put forward a reference framework for association of compounds of communicating locutions with argument linking patterns. It was suggested that a sequence of statements can be made in agent communication, referring to argument networks to reach joint decisions [20]. Agent interaction protocols can be defined on the basis of the utterance of statements, which in turn consists of locutions such as inform, request, reply, query, etc. Protocols and locutions are widely studied in the literature. However, the association of locutions with argumentation during communication has not yet been explicitly established to inform the specification of interaction protocols. The linked argumentation graphs with the identified patterns have the potential to provide a foundation. Overall, this approach is promising in delivering versatile multidisciplinary decision support. Further development will be carried out to fully exploit its potential in more diverse applications.

## Figures and Tables

**Figure 1 healthcare-11-00585-f001:**
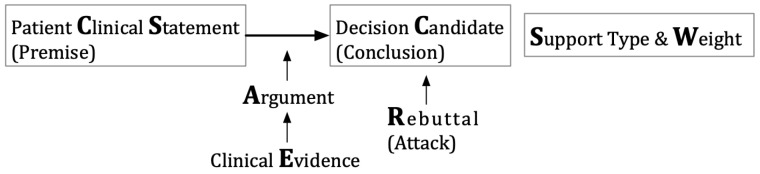
A clinical argumentation scheme.

**Figure 2 healthcare-11-00585-f002:**
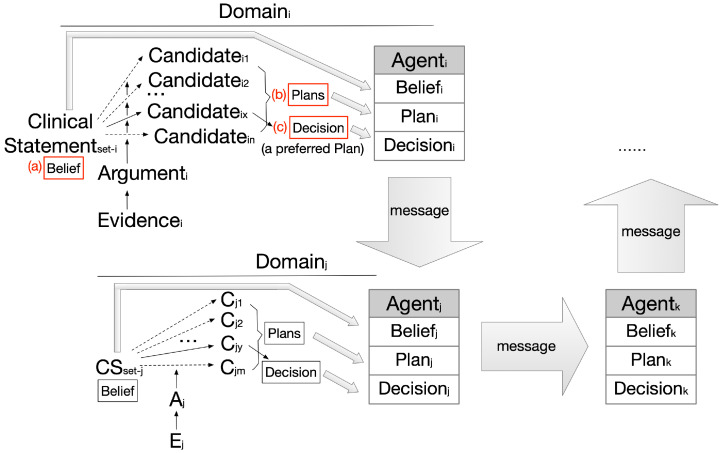
Multiagent argumentation graphs.

**Figure 3 healthcare-11-00585-f003:**
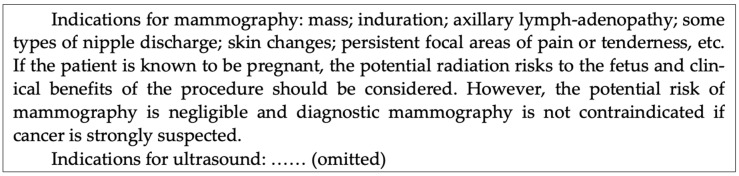
An excerpt from of the evidence for imaging assessment from the Royal College of Surgeons of England guidelines [42].

**Figure 4 healthcare-11-00585-f004:**
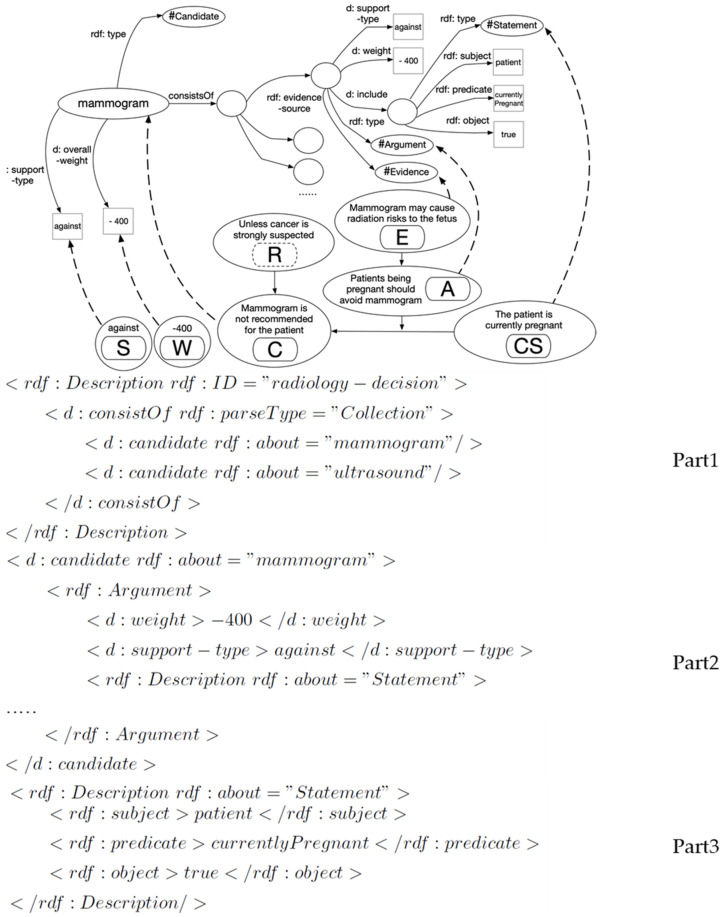
A fragment of the argumentation graph and its RDF representation.

**Figure 5 healthcare-11-00585-f005:**
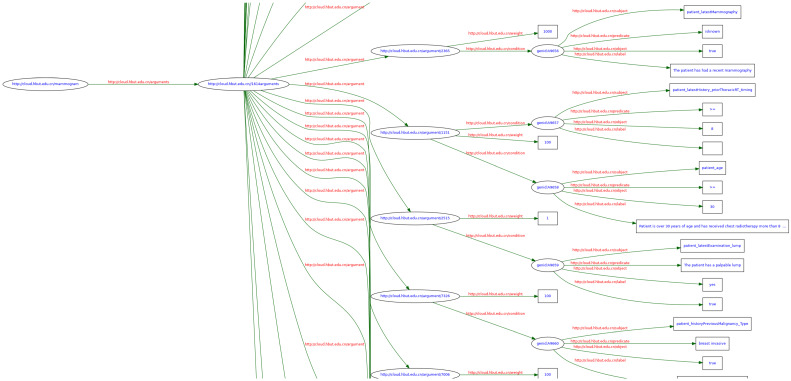
A small portion of the full argumentation graph constructed for the case study.

**Figure 6 healthcare-11-00585-f006:**
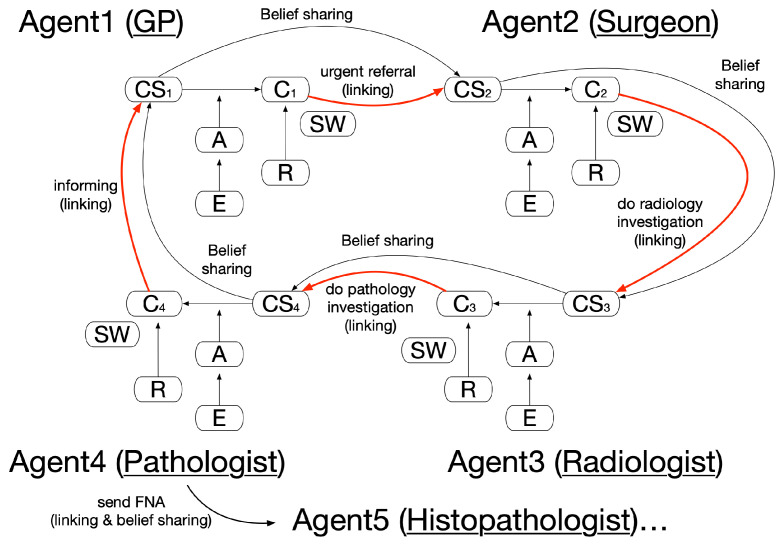
An example of a collaboration pattern of linked argumentation graphs.

**Figure 7 healthcare-11-00585-f007:**
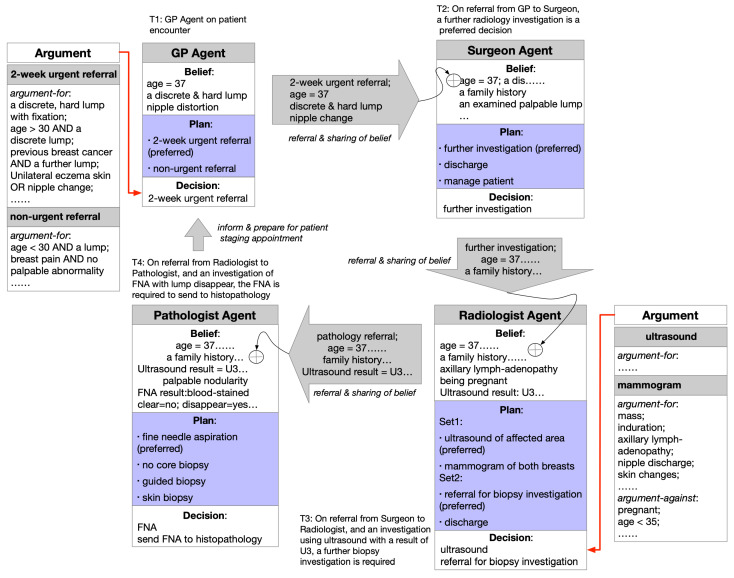
An instantiation of the collaboration pattern for the case study.

**Figure 8 healthcare-11-00585-f008:**
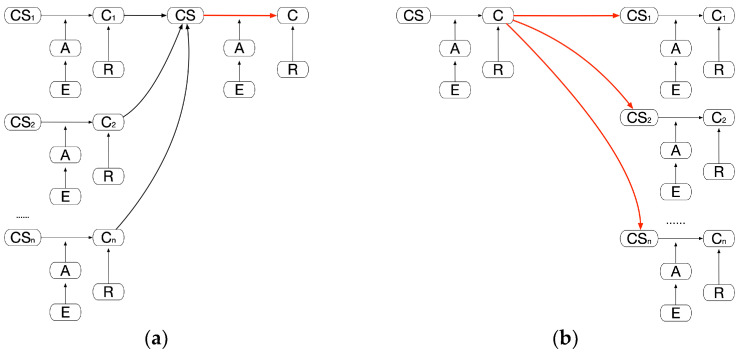
(**a**) “All for one”—“these things are all proved to be true so I could now conclude this”; (**b**) “One for all”—“this big thing is proved to be true so I could now conclude those other pieces”.

**Figure 9 healthcare-11-00585-f009:**
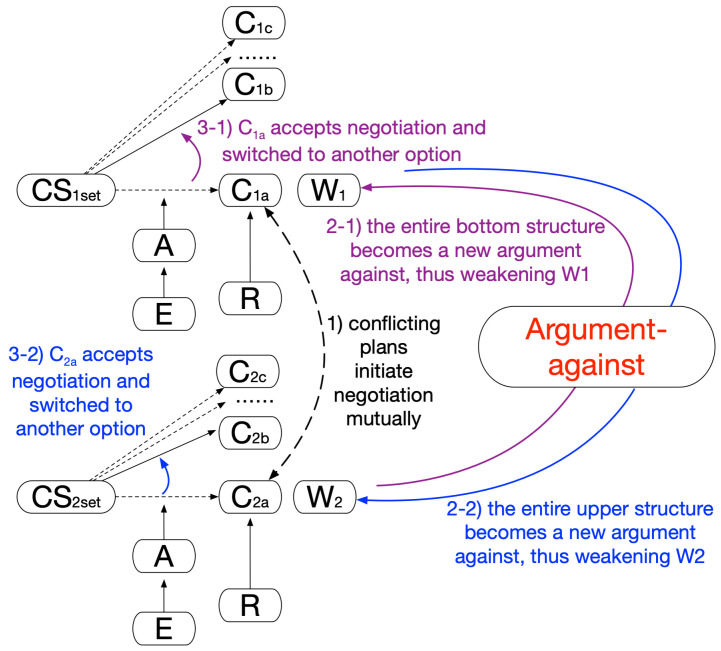
A generic negotiation pattern of linked argumentation graphs.

**Figure 10 healthcare-11-00585-f010:**
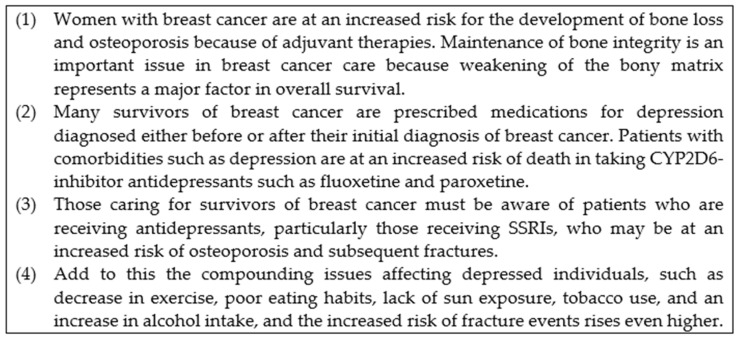
An extraction of evidence [45,46] with minor editing.

**Figure 11 healthcare-11-00585-f011:**
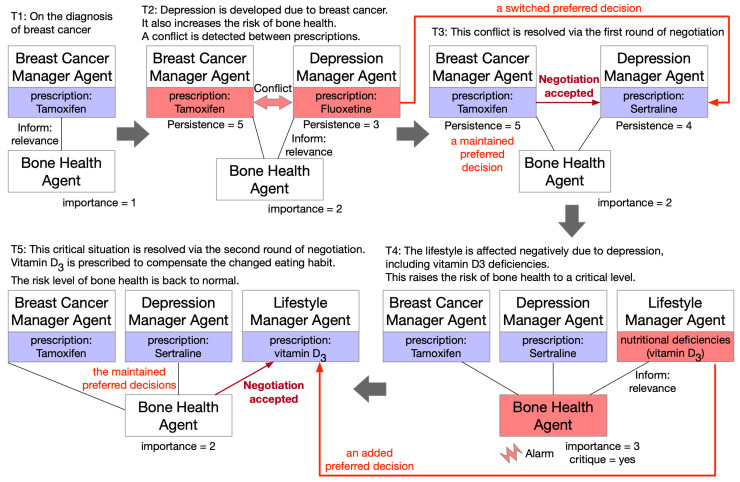
An instantiation of the negotiation pattern with two rounds of negotiation (T2→T3, T4→T5) for the case study.

**Figure 12 healthcare-11-00585-f012:**
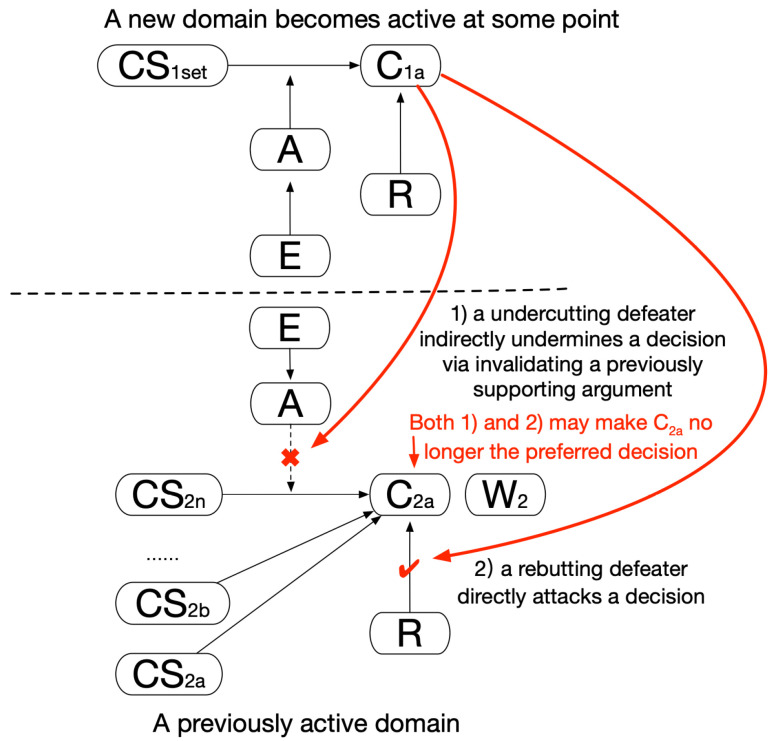
A generic persuasion pattern of linked argumentation graphs.

**Figure 13 healthcare-11-00585-f013:**
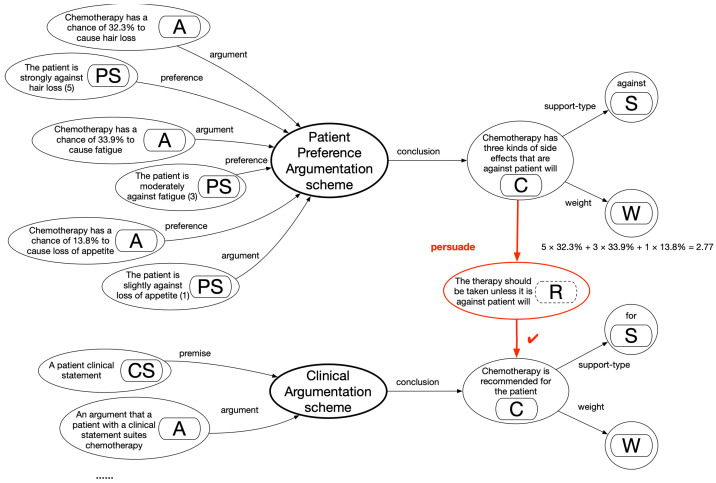
The persuasion pattern applied in the case study, with an example.

**Figure 14 healthcare-11-00585-f014:**
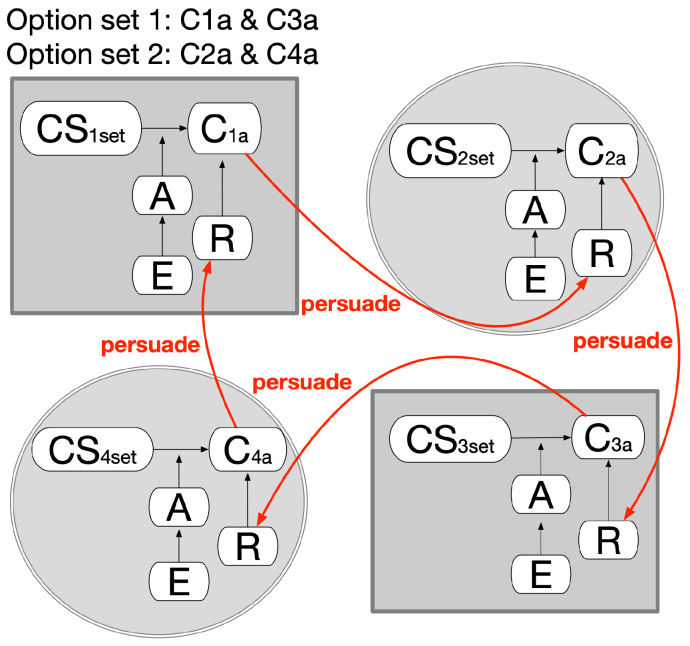
Choosing between different sets of plan options.

**Figure 15 healthcare-11-00585-f015:**
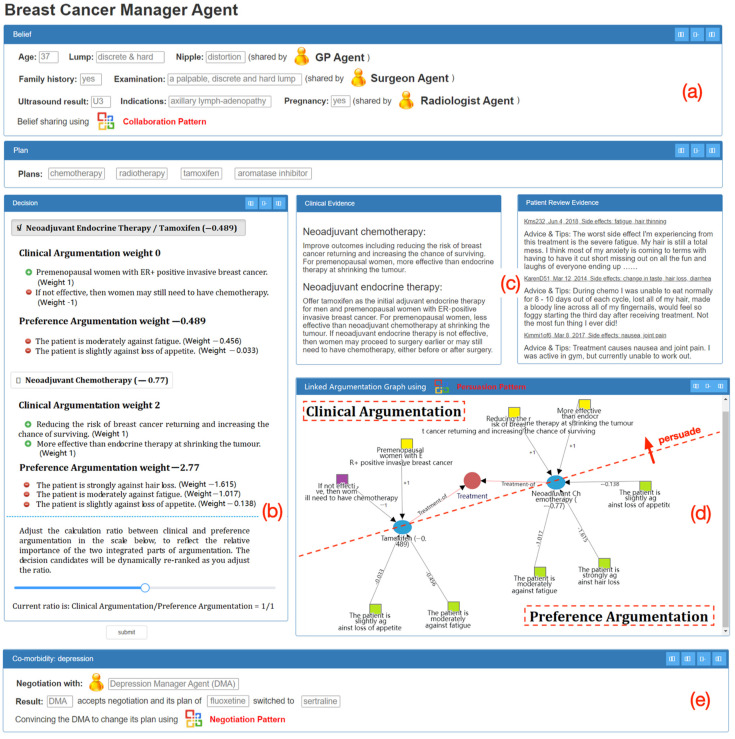
The interface of a prototype multidisciplinary decision support system with the enactments of three types of patterns. (**a**) The belief and plan parts. (**b**) The details on decisions. (**c**) Retrieve clinical evidence as well as patient reviews. (**d**) A graphical view of linked arguments. (**e**) another summary box of the outcome of the negotiation pattern.

**Figure 16 healthcare-11-00585-f016:**
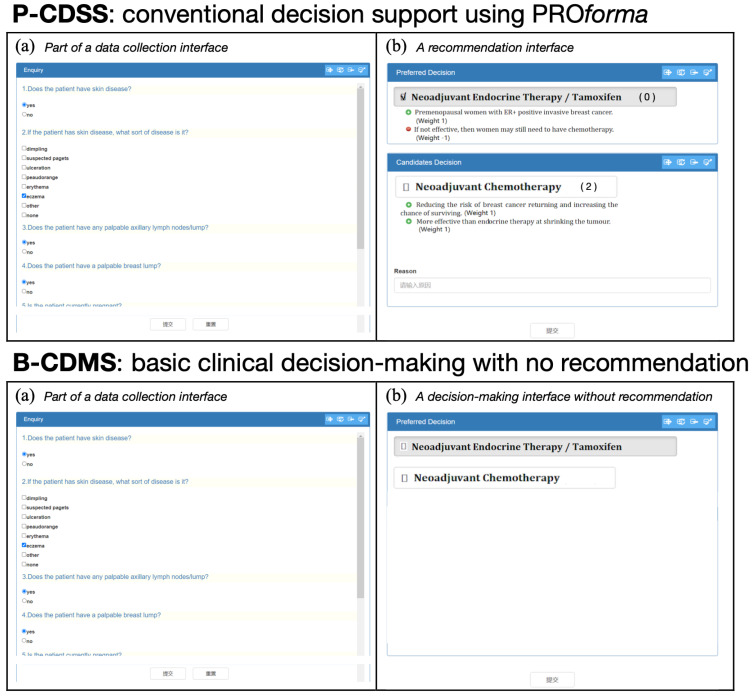
Two additional systems prepared for comparison.

**Figure 17 healthcare-11-00585-f017:**
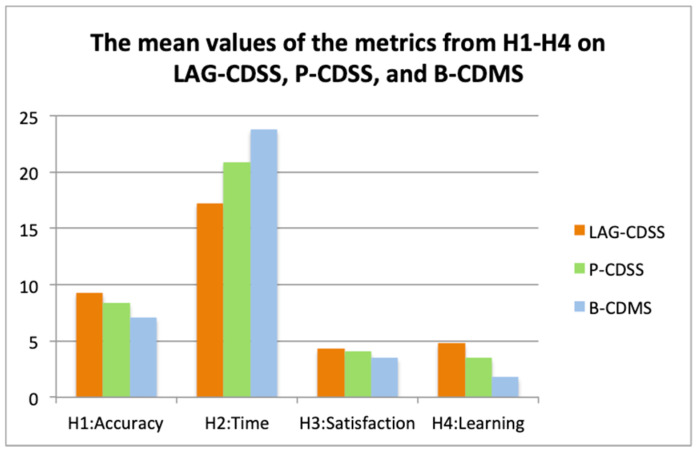
The mean values of the metrics and their comparison.

**Table 1 healthcare-11-00585-t001:** Therapies with top side effects, frequency of occurrence, and sample patient reviews.

	Side Effect	Frequency	Side Effect	Frequency
Chemotherapy	Fatigue	33.9%	Loss of appetite	13.8%
	Hair loss	32.3%	Sore mouth	11.2%
	Nausea	23.2%	Memory problems	9.8%
	Diarrhea	18.5%	Brittle nails	6.2%
	Joint pain	16.9%	Insomnia	3.3%
	Sample Review Evidence			
	Kms232, 4 Jun 2018, Side effects: fatigue, hair thinning Advice & Tips: The worst side effect I’m experiencing from this treatment is the severe fatigue. My hair is still a total mess. I think most of my anxiety is coming to terms with having to have it cut short missing out on all the fun and laughs of everyone ending up ……			
	KarenD51, 12 Mar 2014 Side effects: change in taste, hair loss, diarrhea Advice & Tips: During chemo I was unable to eat normally for 8–10 days out of each cycle, lost all of my hair, made a bloody line across all of my fingernails, would feel so foggy starting the third day after receiving treatment. Not the most fun thing I ever did!			
	Kimmi1of6, 8 March 2017, Side effects: nausea, joint pain Advice & Tips: Treatment causes nausea and joint pain. I was active in gym, but currently unable to work out.			
Radiotherapy	Sore skin	45.6%	Chest pain	6.5%
	Fatigue	21.7%	Lymphedema	4.3%
	Joint pain	8.7%	Hair loss	2.2%
Endocrine therapy— Tamoxifen	Hot flushes	45.5%	Memory problems	9%
	Fatigue	15.2%	Insomnia	6.2%
	Weight gain	12.5%	Loss of appetite	3.3%
Endocrine therapy— Aromatase inhibitors	Joint pain	43.6%	Osteoporosis	10.3%
	Hot flushes	30.8%	Vaginal dryness and bleeding	5.1%
	Insomnia	12.8%	Swelling in feet and legs	2.6%

**Table 2 healthcare-11-00585-t002:** A summary of the metrics for evaluation.

Category	Metric	Description
Productivity	Accuracy	The accuracy of outcomes produced by decision makers collectively as groups.
Process	Time	The total time spent in making decisions, including online/offline interaction among decision makers.
Perception	Satisfaction	The satisfaction of decision makers toward the support received and that of patients toward the generated decision outcomes in alignment with their specific needs.
	Learning	The level of learning involved in the decision-making process and the insights that decision makers can receive from it.

**Table 3 healthcare-11-00585-t003:** The results of hypothesis testing.

Hypothesis	LAG-CDSS		P-CDSS		B-CDMS		*p*-Value1	Confirmation (α = 0.05)	*p*-Value2	Confirmation (α = 0.05)
	Mean	S.D.	Mean	S.D.	Mean	S.D.				
**H1**: Accuracy	9.3	0.675	8.4	0.966	7.1	1.197	0.0133	Yes	0.00004	Yes
**H2**: Time	17.25	1.55	20.85	3.448	23.8	4.373	0.0037	Yes	0.00015	Yes
**H3**: Satisfaction	4.3	0.823	4.1	0.568	3.5	0.527	0.2675	No	0.00927	Yes
**H4**: Learning	4.8	0.422	3.5	0.850	1.8	0.919	0.0002	Yes	<0.00001	Yes

## Data Availability

The data presented in this study are available on request from the corresponding authors.

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
