# Peer review of "Linked Argumentation Graphs for Multidisciplinary Decision Support"

_healthcare, 2023, doi:10.3390/healthcare11040585_

Round 1

Reviewer 1 Report

The paper is devoted to the argumentation-based reasoning in multiagent systems with emphasis in medicine.

While being no expert in the field of argumentation-based reasoning I find it a bit strange that publications cited in the manuscript do not overlap with recent review "A Survey of Toulmin Argumentation Approach for Medical Applications" (https://online-journals.org/index.php/i-joe/article/view/28025) by Fejer et al., 2022. Also authors do not cite review by Carrera and Iglesias "A systematic review of argumentation techniques for multi-agent systems research". Both papers seem to relate to the manuscript topic.

Besides this, I feel I have no right to judge the manuscript's novelty or scientific significance.

Author Response

Thank you very much. The two suggested very relevant works had now been cited as [13] and [36], with references added in the main text of Section 1 and Section 2.3, respectively.

Reviewer 2 Report

The paper entitled “Linked Argumentation Graphs for Multidisciplinary Decision Support” describes, using the example of breast cancer, the decision-making system in medicine, taking into account the opinions of several specialists. The method presented is interesting. The results presented by the Authors lead us to believe that the method presented can assist in the treatment of patients and select optimal or suboptimal therapy in the case of complex issues. For example, in case of interactions between drugs that are recommended by numerous specialists independently. However, in my opinion, the paper needs a few changes before publication. Firstly, the abbreviations RDF and SWRL are not explained. The abbreviation WWAW in line 146 became WWW. 

Secondly, the R in Figure 1 derives from the Rebuttal. This rebuttal was not explained in the text above the figure, where the other letters appearing in the figure were briefly described. Thirdly, Figure 5 is completely unreadable. Whereas figures 16 and 17 are difficult to read. 

At the very end, I would like to ask the authors to explain the Satisfaction survey. The authors acknowledged that the medical student does not reflect well on the mental state of the patient, so the study of this indicator does not, in my opinion, lead to any meaningful conclusions. A medical student who has an idea of treatment methods and their implications, as well as a belief in medicine as a science, will certainly respond differently in such a survey than a patient who is confused (it is, after all, detecting Cancer in him). How is he supposed to be Satisfied with this diagnosis? In my opinion, this indicator proves nothing and should be removed. There is also a strong lack of indication of the novelty of the approach presented.

In my opinion, after these few corrections, the article should be published in the journal.

Author Response

The paper entitled “Linked Argumentation Graphs for Multidisciplinary Decision Support” describes, using the example of breast cancer, the decision-making system in medicine, taking into account the opinions of several specialists. The method presented is interesting. The results presented by the Authors lead us to believe that the method presented can assist in the treatment of patients and select optimal or suboptimal therapy in the case of complex issues. For example, in case of interactions between drugs that are recommended by numerous specialists independently. However, in my opinion, the paper needs a few changes before publication. Firstly, the abbreviations RDF and SWRL are not explained. The abbreviation WWAW in line 146 became WWW. 

Thank you very much. Two abbreviations of RDF and SWRL had now been given their full spellings in Section 2.2. The abbreviation of WWAW in line 146 is short for World Wide Argument Web as a proposed related work. In the following line of 147, WWW is short for World Wide Web. WWAW is intended as a method of interconnecting arguments on WWW in a very large scale.

Secondly, the R in Figure 1 derives from the Rebuttal. This rebuttal was not explained in the text above the figure, where the other letters appearing in the figure were briefly described. Thirdly, Figure 5 is completely unreadable. Whereas figures 16 and 17 are difficult to read. 

We now added a description of the Rebuttal element in the text above Figure 1. Also, we explained further that this element may or may not be explicitly present as part of the scheme, in the following sentence.

Figure 5 had been completely reproduced with much enhanced quality. Figures 16 and 17 were also enlarged with better visibility.

At the very end, I would like to ask the authors to explain the Satisfaction survey. The authors acknowledged that the medical student does not reflect well on the mental state of the patient, so the study of this indicator does not, in my opinion, lead to any meaningful conclusions. A medical student who has an idea of treatment methods and their implications, as well as a belief in medicine as a science, will certainly respond differently in such a survey than a patient who is confused (it is, after all, detecting Cancer in him). How is he supposed to be Satisfied with this diagnosis? In my opinion, this indicator proves nothing and should be removed. There is also a strong lack of indication of the novelty of the approach presented.

Thank you very much. Please let us clarify the issues raised. The Satisfaction metric is defined in Table 2. It measures how decision makers are satisfied with the decision support they received while using the decision support tools, as well as how patients are satisfied with the decision outcomes generated by the tools, in alignment with their specific needs.

In our setting of the experiment, 4 members are grouped together with three playing physicians and one playing a patient. We believe our new method makes a difference from traditional methods in the Satisfaction metric, because it helps with information sharing and adverse effect avoidance during the management of multiple diseases, as brought about by the application of Collaboration and Negotiation Patterns. These are highlighted by the above and bottom parts of the prototype system screenshot in Figure 16. Moreover, the new method supports the integration of patient preference with the power of adjusting objective (clinical) and subjective (patient preference) arguments, as brought about by the application of Persuasion Pattern. This is highlighted by the middle part of the prototype system screenshot in Figure 16, with detailed textual and graphical explanation.

Thus, we believe the analysis reflects, at least in some degree, that participants are more satisfied with the new method in delivering comprehensive decision support or generating patient-oriented decision outcomes. Nevertheless, we acknowledge the limitation of the study in not involving patients in real world. We now mentioned this as part of out future study, following the original text in Section 6.4. We also enlarged Figure 16 to make it clearer for the reviewer to examine our clarification.

In my opinion, after these few corrections, the article should be published in the journal.

Reviewer 3 Report

The paper presents an extended version of our paper published in Proceedings of the 35th IEEE International Symposium on Computer-Based Medical Systems (IEEE CBMS2022). The paper addresses the problem of systematic support for argumentation in communication, among multiple agents. A linked Argumentation Graphs for Multidisciplinary Decision Support is proposed. The paper is well written with detailed discussion on the existing methods covering different categories. The proposed method is supported by experimental results and detailed comparative analysis is presented. It is concluded that the proposed method enhances the performance of the decision support systems. I recommend the paper for publication.    Following minor points need to be addressed in the final of the paper.

The motivation for conducting research needs to be added in the introduction section.

Figure 5 is invisible.

Author Response

Thank you very much. Two major motivations for conducting this research had now been summarized, near the end of the introduction section.

Figure 5 had been completely reproduced with much enhanced quality.

Reviewer 4 Report

The paper is interesting and well written. I have gone through it and find no error in it. I recommend its publication in its present form.

Author Response

Thank you very much for the kind review.

Reviewer 5 Report

The focus of the work presented in the paper " Linked Argumentation Graphs for Multidisciplinary Decision Support" is quite interesting. The authors propose a new approach to Multidisciplinary Decision Support. However, there are many gaps in how techniques were used to test research hypotheses. For a better understanding, it is suggested to make changes such as those mentioned below:

11.     In table 2, you described accuracy as: “The accuracy of outcomes produced by decision makers, collectively as groups.”. However, is not clear, how did the accuracy was calculated?

22.     What Statistical tests were used? I think an ANOVA analysis was used, but   it is not described in the paper.

. 3.   Regardless of the analysis that has been used. It is recommended that the fifteen groups be applied to each of the methods  to do the comparison.

Author Response

The focus of the work presented in the paper " Linked Argumentation Graphs for Multidisciplinary Decision Support" is quite interesting. The authors propose a new approach to Multidisciplinary Decision Support. However, there are many gaps in how techniques were used to test research hypotheses. For a better understanding, it is suggested to make changes such as those mentioned below:

  1. In table 2, you described accuracy as: “The accuracy of outcomes produced by decision makers, collectively as groups.”. However, is not clear, how did the accuracy was calculated?

Thank you very much. It has been described in the last paragraph of Section 6.3, that: “Accuracy was directly measured via the observation of decision-making processes, as the total accurate decision outcomes among 10 patient cases per system.”

  1. What Statistical tests were used? I think an ANOVA analysis was used, but   it is not described in the paper.

Yes, indeed an ANOVA analysis was used, and this had been acknowledged and made clear with much detailed explanation in Section 6.4.

  1. Regardless of the analysis that has been used. It is recommended that the fifteen groups be applied to each of the methods to do the comparison.

We carefully considered this suggestion. Indeed this may support a thorough comparison between the methods. However, we are concerned that if a group has used the more advanced method it may give some indication while working with the other methods afterwards. This could produce unfair measurement and may lead to biased evaluation. Any manner in an attempt to prevent this issue at least would demand more complicated design and management of the experimental settings. As we conducted statistical tests and found no significant difference among groups, the current setting of separate groups using separate methods may be sufficient to produce solid results for analysis. In any sense, we very appreciate this valuable opinion of the reviewer and would definitely think about it to make better experimental design in our future work.

Reviewer 6 Report

This paper is well-written and very interesting for review. 

  Multi-Agent Systems (MASs) is abbreviated in the abstract section. Please only use the MASs introduction section (2nd paragraph) 

This paper is relatively long (however, deep insight) and I would like to request the author to minimize the background and relative works 

Is there a meaningful way to compare the H1-H4 to see how this approach is better than others? 

Author Response

This paper is well-written and very interesting for review. 

Multi-Agent Systems (MASs) is abbreviated in the abstract section. Please only use the MASs introduction section (2nd paragraph) 

Thank you very much, and now only MASs are used in the introduction section and throughout the text.

This paper is relatively long (however, deep insight) and I would like to request the author to minimize the background and relative works 

Thank you again. Three sub-sections summarize the major works of the area: “2.1 Multidisciplinary decision-making in medicine” gives the background of the targeted domain, “2.2 Argumentation theories and representations” provides theoretical background, and “2.3. Multi-agent argumentation” discusses the main techniques of the area. All three parts are highly relevant with our method. As the reviewer noted, they give some deep insight to the research domain. It would be hard to determine which relative works could be deleted without doing harm to the paper. However, we would be very happy to do the necessary edition if more specific advices could be given (one reviewer even suggested to give more references).

Is there a meaningful way to compare the H1-H4 to see how this approach is better than others? 

Yes, the analysis of variance (ANOVA) was used as a statistical testing method for the comparison. This had now been explained with more details, in the beginning part of Section 6.4. 

Reviewer 7 Report

The article is generally well written. The separation between existing approaches and the proposed approach could be made clearer in the first 4 sections where the article seems overly “narrated”. The Maps based approach from Section 5 looks rather straightforward.

Author Response

Thank you very much. The first 4 sections are structured as follows: the first section provides motivation, the second gives background and related works, the third describes the proposed method, and the fourth discusses the different kinds of patterns that could be applied across decision support applications.

The existing approaches and the proposed approach are separated in the second and third sections. Also, the Linked Argumentation Graphs, their patterns, and the instantiations for the case study are visually presented whereas possible, in accompany with the text from Figure 1 to Figure 14.

We very appreciate the suggestion, and would be of more gratitude if more specific advices could be given.

Round 2

Reviewer 5 Report

My recommendations have been answered correctly